# CRISPR/Cas9-mediated generation of biallelic F0 anemonefish (*Amphiprion ocellaris*) mutants

Laurie J. Mitchell[1]*, Valerio Tettamanti[2], Justin S. Rhodes[3], N. Justin Marshall[2], Karen L. Cheney[1], Fabio Cortesi[2]

**1** School of Biological Sciences, The University of Queensland, Brisbane, QLD, Australia, **2** Queensland Brain Institute, The University of Queensland, Brisbane, QLD, Australia, **3** Department of Psychology, Beckman Institute for Advanced Science and Technology, University of Illinois at Urbana, Champaign, Urbana, IL, United States of America

* laurie.mitchell@uqconnect.edu.au

**Data Availability Statement:** All supplementary data and figures can be found in the Supporting Information files. Genetic sequence data is publicly

## Abstract

Genomic manipulation is a useful approach for elucidating the molecular pathways underlying aspects of development, physiology, and behaviour. However, a lack of gene-editing tools appropriated for use in reef fishes has meant the genetic underpinnings for many of their unique traits remain to be investigated. One iconic group of reef fishes ideal for applying this technique are anemonefishes (Amphiprioninae) as they are widely studied for their symbiosis with anemones, sequential hermaphroditism, complex social hierarchies, skin pattern development, and vision, and are raised relatively easily in aquaria. In this study, we developed a gene-editing protocol for applying the CRISPR/Cas9 system in the false clown anemonefish, *Amphiprion ocellaris*. Microinjection of zygotes was used to demonstrate the successful use of our CRISPR/Cas9 approach at two separate target sites: the rhodopsin-like 2B opsin encoding gene (*RH2B*) involved in vision, and Tyrosinase-producing gene (*tyr*) involved in the production of melanin. Analysis of the sequenced target gene regions in *A. ocellaris* embryos showed that uptake was as high as 73.3% of injected embryos. Further analysis of the subcloned mutant gene sequences combined with amplicon shotgun sequencing revealed that our approach had a 75% to 100% efficiency in producing biallelic mutations in F0 *A. ocellaris* embryos. Moreover, we clearly show a loss-of-function in *tyr* mutant embryos which exhibited typical hypomelanistic phenotypes. This protocol is intended as a useful starting point to further explore the potential application of CRISPR/ Cas9 in *A. ocellaris*, as a platform for studying gene function in anemonefishes and other reef fishes.

## Introduction

Targeted genome modification (i.e., reverse genetics) is an elegant approach for directly attributing genotype with phenotype but has been limited in non-model organisms owing to a lack of high-quality assembled genomes, affordable technologies, and species-specific protocols.

available for download from The University of Queensland's Resource Data Manager https://doi.org/10.48610/ddf0baa.

**Funding:** This research was funded by an Australian Research Council Discovery Project (DP18012363) awarded to N.J.M. and F.C. K.L.C was furthermore supported by an ARC Future Fellowship (FT190100313) and F.C. was supported by an ARC DECRA (DE200100620) and a University of Queensland Development Fellowship. https://www.arc.gov.au/ The funders had no role in study design, data collection and analysis, decision to publish, or preparation of the manuscript.

**Competing interests:** The authors have declared that no competing interests exist.

Modern gene-editing tools such as the clustered-regularly-interspaced-short-palindromic-repeat (CRISPR/Cas9) system enables precise targeted gene-editing, where a synthetic guide RNA (sgRNA) directs the cutting activity of Cas9 protein to produce a double strand break at a genetic location of interest. Subsequent error prone DNA repair by non-homologous end joining (NHEJ) often leaves insertions and/or deletions (indels), which may induce a frame-shift and potential loss of gene function [1]. The injection of sgRNA fused with Cas9 protein has proven to be an effective tool for precise genome editing at target gene sequences in the cell lines of numerous species including many teleost fishes such as zebrafish (*Danio rerio*) [2], Nile tilapia (*Oreochromis niloticus*) [3,4], medaka (*Oryzias latipes*) [5], Atlantic salmon (*Salmo salar*) [6], killifish (spp.) [7,8], pufferfish (*Takifugu rubribes*) [9,10], and red sea bream (*Pagrus major*) [11]. However, the CRISPR/Cas9 system has yet to be applied to coral reef fishes, a highly diverse assemblage of species with a unique life history and biological adaptations suited for survival in their reef environment (e.g., a pelagic larval stage, demersal spawning, and parental behaviour) [12–14] but make them incompatible with standard CRISPR protocols used on most teleosts. Thus, requiring the development of a new approach.

One such group of reef fishes are anemonefishes (subfamily, Amphiprioninae), an iconic group that shelter in certain species of sea anemones [15], and are sequential hermaphrodites [16,17] that live in strict social hierarchies determined by body size [18]. The fascinating aspects of anemonefish biology has led to their use in multiple areas of research including for studying the physiological responses of reef fishes to the effects of climate change [19–21], the hormonal pathways that regulate sex change [22,23] and parental behaviour [24–26], and the physiological adaptations for group-living [18,27]. Moreover, anemonefishes are being used to understand the visual capabilities of fish [28,29] and evolution of skin colour diversity [30–32] in reef fishes. However, despite the wide-reaching applications of anemonefish research, the genetic basis for many of their traits remain to be empirically investigated. Consequently, anemonefish studies have been limited to correlative findings from comparative transcriptomics [30–32] and/or indirect comparisons by using reverse genetics/testing genetic elements of interest in pre-established models such as zebrafish [32]. Recently, the release of assembled genomes for multiple anemonefish species [33–35] has made it feasible to apply the CRISPR/Cas9 system to conduct genome modification in anemonefishes.

Producing biallelic knockout animals within the first generation (F0) is often desirable in species with long generation times where the establishment of stable transgenic lines might take several years. This is true for anemonefishes which have a relatively long development time till reproduction (~12–18 months), and therefore, are poorly suited for studies that rely on multigenerational breeding schemes to generate results. Thus, a well-designed protocol for the efficient delivery of sgRNA/Cas9 to completely knockout gene function is needed. To achieve this, careful species-specific considerations must be made for sgRNA design, dose toxicity, construct delivery parameters (i.e., air pressure, needle dimensions), and egg/embryo-care both during microinjection and incubation [10]. Inherent challenges specific to gene-editing anemonefishes and many other demersal spawning reef fishes include the injection and/or care of their substrate-attaching eggs [36] and pelagic larval stage [37]. Therefore, a protocol for performing CRISPR/Cas9-mediated genome editing in anemonefishes would be highly beneficial for diverse areas of research to directly test candidate genes facilitating e.g., sex change [23], colour vision [29] and skin pattern development [32].

In this study, we describe a protocol for performing CRISPR-Cas9 in the false clown anemonefish, *Amphiprion ocellaris*, an ideal species for gene-editing due to the public availability of its long-read assembled genome [33], its relative ease of being cultured in captivity [38] and being the most widely studied anemonefish species [39]. This has led the community to work on adapting a CRISPR/Cas9 approach simultaneously [40]. To demonstrate our protocol, we

report on its efficacy in producing biallelic knockouts in F0 *A. ocellaris* embryos. Newly fertilised embryos were injected with a construct of synthetic guide RNA and recombinant Cas9 protein that separately targeted two genes: The rhodopsin-like 2B opsin gene (*RH2B*) encoding a mid-wavelength-sensitive visual pigment [41], and the Tyrosinase encoding gene (*tyr*) involved in the initial step of melanin production [42]. Moreover, genomic sequencing and skin (melanism) phenotypes revealed in many individuals a complete loss of gene function. We hope this protocol provides a useful resource for future gene-editing experiments involving anemonefishes and other demersal spawning reef fishes.

## Materials and methods

### Care and culturing of *A. ocellaris*

Captive-bred pairs of *A. ocellaris* purchased from a local commercial breeder (Clownfish Haven, Brisbane Australia) were housed in recirculating aquaria at The Institute for Molecular Bioscience at The University of Queensland, Australia. Experiments were conducted in accordance with Animal Ethics Committee guidelines and governmental regulations (AEC approval no. QBI/304/16; Australian Government Department of Agriculture permit no. 2019/066; UQ Institutional Biosafety approval no. IBC/1085/QBI/2017). Individual aquaria (95 L) contained a single terracotta pot (27 cm diameter) that provided a shelter and egg-laying structure for brood-stock fish. Spawning usually occurred during the late-afternoon to evening (15:00–18:00), which was preceded by a fully protruded ovipositor and behaviours that included surface cleaning and ventral rubbing on pot surfaces. Eggs laid by the female were adhered to the pot and subsequently fertilised by the male (Fig 1). Because injected eggs are rejected by parents after being returned, we incubated the eggs in an isolated tank (36 L) which contained heated (26˚C) marine water (1.025 sg) dosed with methylene blue (0.7 mL, Aquasonic), and kept aerated using a wooden air diffuser (Red Sea). Dead eggs/embryos were removed daily to minimise the risk of fungal or other disease outbreaks.

Six to seven days post-fertilisation, the eyes of embryos visibly silvered, and they were ready to hatch. Because larvae in our system (both mutant and wildtype) often struggled to hatch properly despite being provided optimal external conditions (e.g., no-light, warmth, water motion), we resorted to using a non-standard approach, where larvae were manually hatched. Eggs which contained larvae were viewed under a microscope while immersed, and a small

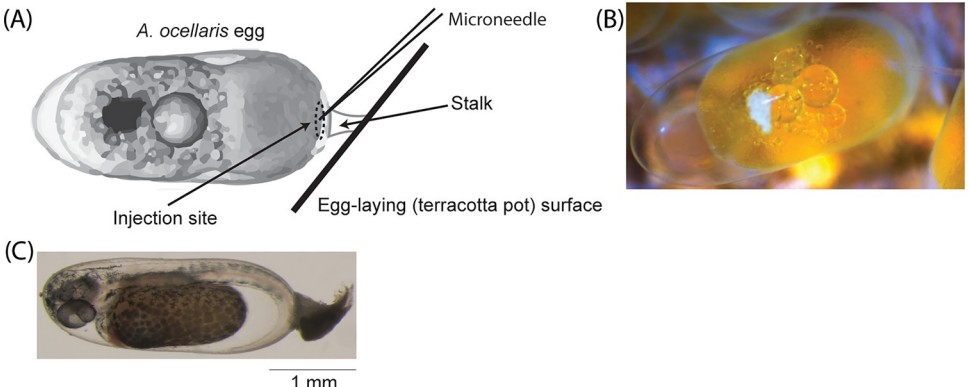

**Fig 1. Egg microinjection site and embryo appearance.** (A) Illustration depicting an *Amphiprion ocellaris* egg (< 1-hour post-fertilisation) with demarcated injection site at the animal pole. (B) Brightfield micrograph of a live *A. ocellaris* egg injected by a microneedle with released fluid marked by red-fluorescent dye. (C) Wild-type *A. ocellaris* embryo at 64–88 hours post-fertilisation with formed eyes and pigmentation.

pair of dissection scissors were used to make an incision near the base of the egg on its sub-strate-attaching side, and then a pair of fine-tipped forceps were used to gently pry the chorion apart to produce a large enough hole for the larva to emerge. Free-swimming larvae were then immediately transferred to a grow-out tank (35x20x35 cm) and raised following a standard anemonefish rearing approach [for more details and alternative protocols see 43]. Larvae were kept in an aquarium with sides wrapped in black plastic that eliminated all horizontal light to prevent bodily damage from repeated swimming into the tank walls. Tank water was kept cir-culated using a low flowrate air pump with stone. Live rotifers (*Brachionus* spp.) were intro-duced at a high dosage (~10 rotifers/mL) as a food source, along with microalgae (*Nannochloropsis* spp.) which tinted the water green. 24-hrs post-hatch (i.e., 1 dph), a very dim overhead (single blue LED strip) light on a 12:12 hour timer was introduced to encourage feed-ing while not stressing the larvae. Rotifer density was maintained till 6 to 7 dph, after which freshly (24-hr old) hatched nauplii of *Artemia* spp. were introduced. By 10 dph, the diet of ane-monefish larvae was fully transitioned to exclusively *Artemia* (~3 nauplii/mL). An air-sponge filter was installed 10 dph to control ammonia levels, and overhead lighting was changed to a slightly brighter white light. An artificial diet of pellets (75–250 μm) was introduced 14 dph, which coincided with the completion of metamorphosis. Juveniles (~30 dph) were transitioned to larger pellets (500–800 μm, Ocean Nutrition), by which point fish were approximately 3.0 cm in standard length.

## Design and in-vitro testing of sgRNAs

To trial the application of the CRISPR-Cas9 system in anemonefishes, we designed four and two sgRNAs that targeted *A. ocellaris RH2B* and *tyr* genes, respectively (Fig 2A and 2B). The gene sequence for *A. ocellaris RH2B* was obtained from a previous study [29], and the same approach described by Mitchell et al. 2021 was used to identify the *tyr* gene sequence in the *A. ocellaris* genome [33]. All gene sequences were viewed in Geneious Prime (v.2019.2.3, https://www.geneious.com/), and the "Find CRISPR Sites. . ." function was used to screen suitable sgRNA sequences with search parameters that included a target sequence length of 19-bp or 20-bp, an 'NGG' protospacer-adjacent-motif (PAM) site located on the 3' end of the target sequence (see Supporting Information S1 File for a list of sgRNA sequences). All selected target sequences were scored for their off-target activity compared against the *A. ocellaris* genome using an inbuilt scoring algorithm implemented in Geneious and originally designed by Hsu et al. (2013) [44]. Each off-target site is given a score based on how similar it is to the original CRISPR site and where any mismatches occur (i.e., mismatches near the PAM site will affect binding more than those further away from the PAM site). A higher score for an off-target site indicates a higher similarity to the original CRISPR site, and a higher likelihood of the sgRNA/Cas9 binding to the off target. The overall specificity score for a CRISPR site is calculated as 100% minus a weighted sum of off-target scores in the target genome. Thus, a high specificity score indicates a more ideal CRISPR site with few or weak potential off targets. We screened and selected sgRNAs with no major off-target sites (overall specificity score ≥90%). Both the sgRNAs and purified Cas9 protein fused with nuclear-localisation-signal (NLS) used in this study were purchased from Invitrogen (catalogue no. A35534, A36498; https://www.thermofisher.com/). One forward-directed cutting sgRNA on the *RH2B* gene targeted a sequence on Exon 4 immediately upstream (18-bp) of the conserved chromophore binding site Lys296 [45], where a frameshift would prevent the formation of a functional visual pig-ment. To assess cutting activity at other *RH2B* sites, we selected three additional target sequences, including one on Exon 1, and two on Exon 5 (i.e., downstream of Lys296). Two sgRNAs targeted sites on Exon 2 of the *tyr* gene, a location adequately upstream where reading

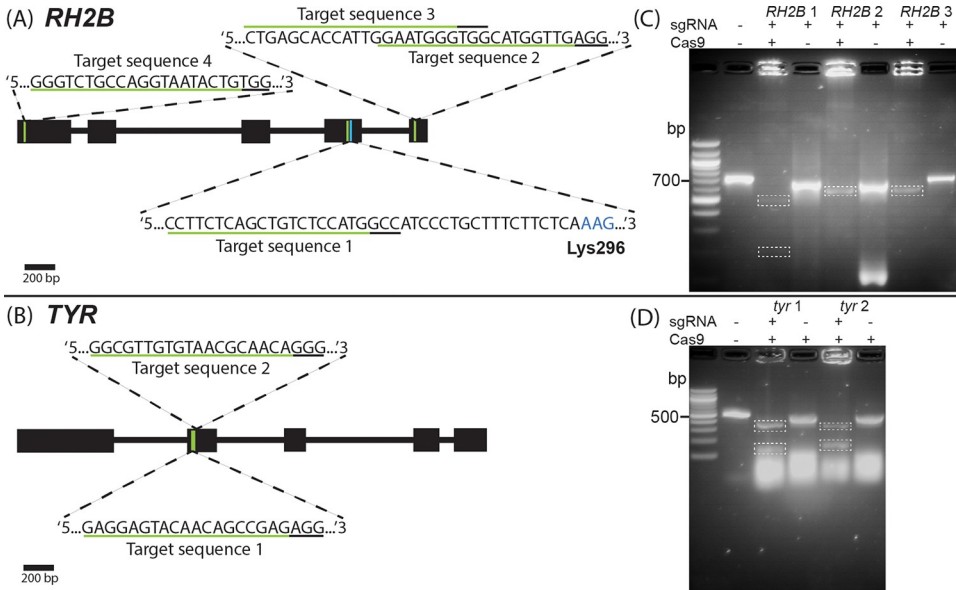

**Fig 2. Targeted gene regions and in-vitro cutting assay.** Sites and sequences targeted by sgRNA designed for the (A) *RH2B* and (B) *tyr* genes in *Amphiprion ocellaris*. Expanded regions show the target sequence (underlined in green) and 'NGG' PAM (underlined in black) for each sgRNA. For Exon 4 of *RH2B*, the Lys296 chromophore binding site (coloured blue) is also depicted down-stream of target sequence 1. Gel images to the right of each gene illustration depict DNA fragments size when amplicons that contained target (C) *RH2B* and (D) *tyr* gene regions were incubated (in-vitro) with (+) or without (-) Cas9 protein and sgRNA. Dotted boxes highlight cut DNA fragments.

frame shifts produced by indel mutations would more likely knockout gene function, while being far enough downstream to reduce the likelihood of alternative transcription start sites being utilised. The cutting activity of our sgRNAs with Cas9 were initially assessed *in-vitro* by incubating PCR amplicons of targeted gene regions with or without sgRNA and/or Cas9 and comparing fragment length via gel electrophoresis (see Supporting Information S2 File for full details on PCR routine, and in-vitro assay reagent quantities and incubation parameters) (Fig 2C and 2D).

## Microinjection delivery of CRISPR-constructs

The clutches were collected 10–15 minutes post-fertilisation for CRISPR-construct delivery to ensure adequate fertilisation of eggs but before the first cell division had occurred 60–90 min post-fertilisation [46]. Both before and during injecting, pots containing egg clutches were broken apart into multiple shards (~2.0x4.0 cm) using a hammer and chisel. The shards were then placed in a petri dish and partially submerged in Yamamoto's ringer's solution [47] (see Supporting Information S3 File) to alleviate osmotic stress associated with injection [10]. Eggs were viewed under a dissection microscope (3.5x magnification) and microinjected directly into the animal pole (Fig 1A and 1B) at a 45˚ angle with a pulled borosilicate glass pipette (Harvard Apparatus: 1.0x0.58x100 mm) fitted on a pneumatic injector unit (Narishige IM- 400) and micromanipulator (Marzhauser MM3301R). Injector pressure settings were configured to deliver a 1 nL dose of a mixture per egg. Our initial mixture contained sgRNA (200 ng/μL, 13.8 μM), Cas9 protein (500 ng/μL, 12.3 μM) and KCl (300 μM), that was suspended by slowly pipetting up-and-down in a 10 μL stock-solution containing 5.5 μL RNAse free $H_2O$ and incubated at 37˚C for 10 minutes to form a sgRNA/Cas9 construct then stored on ice, 20–30 min before injections started. Both the inclusion of KCl solution to aid in sgRNA/Cas9 mix

solubility, and incubation step were adapted from Burger et al. (2016) [48]. 2 μL of the solution was then backloaded into a microneedle immediately before injection (see Supporting Information S4 File for details on microneedle dimensions and injector pressure settings). Injecting ceased when the chorion had become too thick to penetrate (~40–50 minutes post-fertilisation). To assess the mortality attributed to toxicity of the injection dosage and damage induced loss, the survival rate of CRISPR-Cas9 injected eggs were compared to controls, including: *1*) eggs injected with a mixture containing no Cas9 (replaced with water), *2*) non-injected eggs, and *3*) a mixture containing diluted sgRNA (5 μM) and Cas9 protein (5 μM).

To control for differences in individual user, we had multiple personnel perform injections across clutches. Survival rates were calculated as the proportion of live embryos (Fig 1C) at collection relative to the number of embryos per treatment at <1-hour post-fertilisation (hpf).

### Genotype and phenotype analysis of mutants

Treatment and control embryos were collected 64–88 hours post-fertilisation when eyes were clearly visible (Fig 1C). Tissue samples were taken as fin-clips from juveniles at about three months post-hatch. DNA was extracted from embryos and fin-clips using a DNeasy Blood & Tissue kit (Qiagen catalogue no. 69504), as per the manufacturer's protocol. The concentration and purity of the extracted DNA was first tested via Nanodrop (IMPLEN N60) and then PCR-amplified using primers flanking the targeted gene location (see Supporting Information S2 File for primer sequences). Sanger sequencing of PCR amplicons was outsourced to AGRF (https://www.agrf.org.au/) and positive mutants were detected by mapping their sequences against the respective gene in Geneious. Because all positive mutants had a degree of mosaicism, we identified the full range of mutations by subcloning the PCR products of four *RH2B* (clutch no. 3) and four *tyr* (clutch no. 12) mutant embryos from clutches with high somatic activity using the Invitrogen TOPO TA kit according to the manufactures protocol (Invitrogen catalogue no. K4575J10), and Sanger sequenced the extracted plasmids from 6–10 colonies per sample. This process was also performed using fin-clips taken from three-month-old *RH2B* mutant juveniles (n = 3) from clutch no. 16.

To further analyse the mosaicism of our mutagenesis approach, we submitted two samples per target gene for next generation shotgun amplicon sequencing (NGS) to Novogene (https://en.novogene.com/) using Illumina NovaSeq paired-end sequencing with insert sizes of 150bp for *RH2B* (1 Gbp) and 250bp for *tyr* (1 Mbp). Raw reads were processed in Geneious by first trimming adapters and low-quality bases (phred scores <20) from the end of reads using the 'BBDuk' plugin (v.38.84; https://www.geneious.com/plugins/bbduk/), and then merged paired reads using the tool 'BBMerge'. Next, we used the 'Analyse CRISPR Editing Results' tool in Geneious which mapped merged reads to an unedited reference sequence (50bp sequence forward and reverse of the CRISPR site), then collapsed identical variants (≥0.04% minimum variant frequency) and returned a number of reads as a percentage of the total mapped reads.

Brightfield micrographs were taken (Nikon SMZ800N) of individual *tyr* mutant embryos and a wildtype embryo for comparison.

## Results and discussion

### sgRNA *in-vitro* assay

An in-vitro assessment of sgRNA cutting activity was conducted to verify the integrity and viability of our sgRNA designs with target sites located on either *A. ocellaris RH2B* opsin gene (Fig 2A) or *tyr* gene (Fig 2B). All five selected sgRNAs exhibited positive cutting activity after incubation with amplicons that encompassed the target regions (Fig 2C and 2D). Cutting

**Table 1. Clutch survival and mutation rates for *RH2B* targeted injection rounds.** Survival rates of embryos injected with sgRNA/Cas9 (treatment) or sgRNA-only (positive control) targeting *RH2B*, and non-injected (negative control) embryos at time of collection (64–88 hpf), number of genotyped embryos, and mutation rate per clutch and target sequence. Note: Clutches 8–11 (sgRNA 4) were injected with a lower concentration of sgRNA/Cas9.

| Clutch no. | 1 | 2 | 3 | 4 | 5 | 6 | 7 | 8 | 9 | 10 | 11 |
|---|---|---|---|---|---|---|---|---|---|---|---|
| *RH2B* sgRNA | 1 | 1 | 1 | 2 | 2 | 3 | 3 | 4 | 4 | 4 | 4 |
| Injected survival | 11/22 | 14/48 | 13/48 | 22/75 | 7/56 | 17/50 | 16/45 | 75/147 | 64/277 | 43/141 | 102/340 |
| 64–88 hpf | 22.0% | 29.2% | 27.1% | 29.3% | 12.5% | 34.0% | 35.6% | 48.3% | 23.1% | 30.5% | 30.0% |
| No. of genotyped (out of injected) | 11/11 | 13/14 | 13/13 | 22/22 | 6/7 | 17/17 | 9/16 | 15/75 | - | - | 7/19* |
| Positive mutants | 1/11 9.1% | 3/13 23.1% | 4/13 30.8% | 0/22 0% | 1/6 16.7% | 8/17 47.1% | 3/9 33.3% | 11/15 73.3% | - | - | 4/7* |
| Non-injected survival rate 64–88 hpf | 25/59 42.4% | 40/50 80.0% | 25/42 59.5% | 14/26 53.8% | 16/52 30.8% | 44/47 93.6% | 51/61 83.6% | 47/80 58.8% | 71/92 77.2% | - | - |
| Injected (sgRNA-only) survival 64–88 hpf | - | - | - | - | 4/10 40.0% | 5/12 41.7% | - | - | - | - | - |

hpf, hours post-fertilisation.

*proportion of positive mutants identified by fin-clip extractions taken from juveniles.

activity indicated the sgRNA designs were suitable for in-vivo trials. No cutting activity was observed when amplicons were incubated without sgRNA (for *tyr*) or Cas9 (for *RH2B*).

## Survival and mutation rate

Overall, negative control or non-injected clutch survival ranged between 25.7% to 93.6% (mean ± sd = 62.9 ± 19.0%) and was consistently higher than survival of sgRNA/Cas9 injected embryos, which ranged from 12.5% to 48.3% in *RH2B* targeted embryos (29.2 ± 9.0%), and 16.3% to 27.7% in *tyr* targeted embryos (22.2 ± 4.8%) (Tables 1 and 2). However, inter-clutch survivability was overall highly variable, a possible consequence of variable broodstock quality and/or experience levels in spawning. Survival of positive control (sgRNA-only injected) embryos ranged between 26.1% to 73.7% (45.4 ± 17.4%) (Tables 1 and 2), and no clear improvement was detected when eggs were injected with a >50% reduced concentration of sgRNA/Cas9 (33.0 ± 10.8%) (in clutches 8–11; Table 1). These observed differences in survival between the injected treatments and (non-injected) control embryos, indicated that physical trauma from the injection process was most likely the main contributor to mortality observed in injected embryos. A reduction in needle tip-size (<15 μm) may help lower mortality;

**Table 2. Clutch survival and mutation rates for *tyr* targeted injection rounds.** Survival rates of embryos injected with sgRNA/Cas9 or sgRNA-only targeting *tyr*, and non-injected embryos at time of collection (64–88 hpf), number of genotyped embryos, and mutation rate per clutch and target sequence.

| Clutch no. | 12 | 13 | 14 | 15 | 16 |
|---|---|---|---|---|---|
| *tyr* sgRNA | 1 | 1 | 2 | 2 | 2 |
| Injected survival | 14/86 | 74/267 | 9/47 | 39/148 | 27/126 |
| 64–88 hpf | 16.3% | 27.7% | 19.1% | 26.4% | 21.4% |
| No. of genotyped (out of injected) | 13/14 | 74/74 | 8/9 | 39/39 | 27/27 |
| Positive mutants | 7/13 53.8% | 9/74* 12.2%* | 2/8 25.0% | 7/39* 17.9%* | 12/27* 44.4% |
| Non-injected survival rate 64–88 hpf | 37/45 82.2% | 19/74 25.7% | 3/10 30.0% | 52/67 77.6% | 57/67 85.1% |
| Injected (sgRNA-only) survival 64–88 hpf | - | - | - | 12/46 26.1% | 14/19 73.7% |

hpf, hours post-fertilisation

*proportion of positive mutants identified solely by hypomelanistic phenotype out of all surviving embryos.

however, in our experience thinner needles exhibited excessive bending when attempting to penetrate the thick chorion of anemonefish eggs. Only needles with a relatively short-taper and broad tip (i.e., stubby profile) were usable for injections. Natural thickening of the chorion peaked at 30–40 minutes post-fertilisation (about 50–60 minutes preceding the first cell division) and prohibited further injecting regardless of needle size.

Examination of the target gene sequences of injected embryos showed highly variable mutation rates that ranged from 0% to 73.3% for *RH2B* (n = 10 clutches; Table 1), and 12.2% to 53.8% for *tyr* (n = 5 clutches; Table 2). In *RH2B* targeted fishes raised till the juvenile-stage (clutch 11; Table 1), we found a relatively high mutation rate of 57.1%. We also found that lowering the injected sgRNA/Cas9 concentration (<50%) had no apparent impact on mutation rate (clutch 8 = 33.3%, clutch 11 = 57.1%; Table 1). To achieve higher mutation rates, we suggest a couple alternative options such as by improving the accuracy of injecting the animal pole by delaying injection until the formation and visible swelling of the blastodisc (~40–50 minutes post-fertilisation) that precedes the first cell division; however, this severely limits the number of injectable eggs due to thickening of the chorion. Alternatively, the substitution of Cas9 protein with Cas9 mRNA may circumvent the need for direct delivery into the nucleus and permit injection elsewhere (e.g., in the yolk). Although Cas9 protein has been associated with a higher efficiency of mutagenesis than Cas9 mRNA [49], the relatively long-lived (~90 minutes) single cell stage of the *A. ocellaris* zygote [46] would likely permit adequate time for migration into the nucleus and translation processes. The incorporation of NLS-fused Cas9 mRNA could also help compensate for differences in uptake efficiency [50].

## Genotype analysis of mutants

Analysis of the subcloned sequences of *RH2B* (clutch 3, *RH2B* 1; Fig 3A) and *tyr* (clutch 12, *tyr* 1; Fig 3B) mutant *A. ocellaris* embryos, revealed that our approach was successful in producing biallelic mutations in seven out of the eight embryos; only one *tyr* mutant retained a wildtype allele. This high (75% to 100%) efficiency in inducing biallelic mutations in F0 *A. ocellaris* proves promising for the use of reverse genetics in animals with long generation times (12–18 months in the case of anemonefishes) [51], allowing experiments to start while waiting for stable homozygous-lines to be established. Although verifying germline transmission in F0 brood-stock will be required for long-term, inter-generational studies.

A total of 24 and 11 distinct mutations were found in *RH2B* mutants (Figs 3A and 4A) and in *tyr* mutants (Fig 3B), respectively. Although most mutations were detected by both the sequencing of subcloned colonies and NGS (see Supporting Information S5 File for full details on all variants detected by NGS), the greater sampling depth of the latter (total no. of reads: *RH2B*-M1 = 3649143, *RH2B*-M4 = 3518312, *tyr*-M2 = 83196, *tyr*-M4 = 664560) revealed additional mutations in *RH2B*-M1 (n = 4), *RH2B*-M4 (n = 2), *tyr*-M2 (n = 1), and *tyr*-M4 (n = 4). Most mutations were in the form of deletions that ranged in length between 1 – 43bp, while fewer insertions ranged from 1–10 bp. An extremely large deletion of 449bp was detected in *RH2B*-M5 and *RH2B*-M6 (Fig 4A). Mutations were situated (4 – 14bp) upstream ('5) of their respective PAM sequence, a proximity and location typically reported for Cas9 cutting activity [52] (Fig 3A and 3B). Exceptions included deletions starting at the PAM in *tyr*-M2 and *tyr*-M3 (-7bp), and that spanned regions both up- and down-stream of the PAM in *RH2B*-M4 (-43bp), *RH2B*-M5 and *RH2B*-M6 (-449bp). The most frequent mutations found in multiple *RH2B* mutants included a 5bp deletion (10bp upstream of PAM) and a 2bp deletion (14bp upstream of PAM) (Fig 3A), while the most common mutations across *tyr* mutants were a 1bp deletion (4bp upstream of PAM) and a 7bp deletion (starting at PAM) (Fig 3B). Both *RH2B* (Figs 3A and 4A) and *tyr* (Fig 3B) mutant embryos had between two to seven distinct

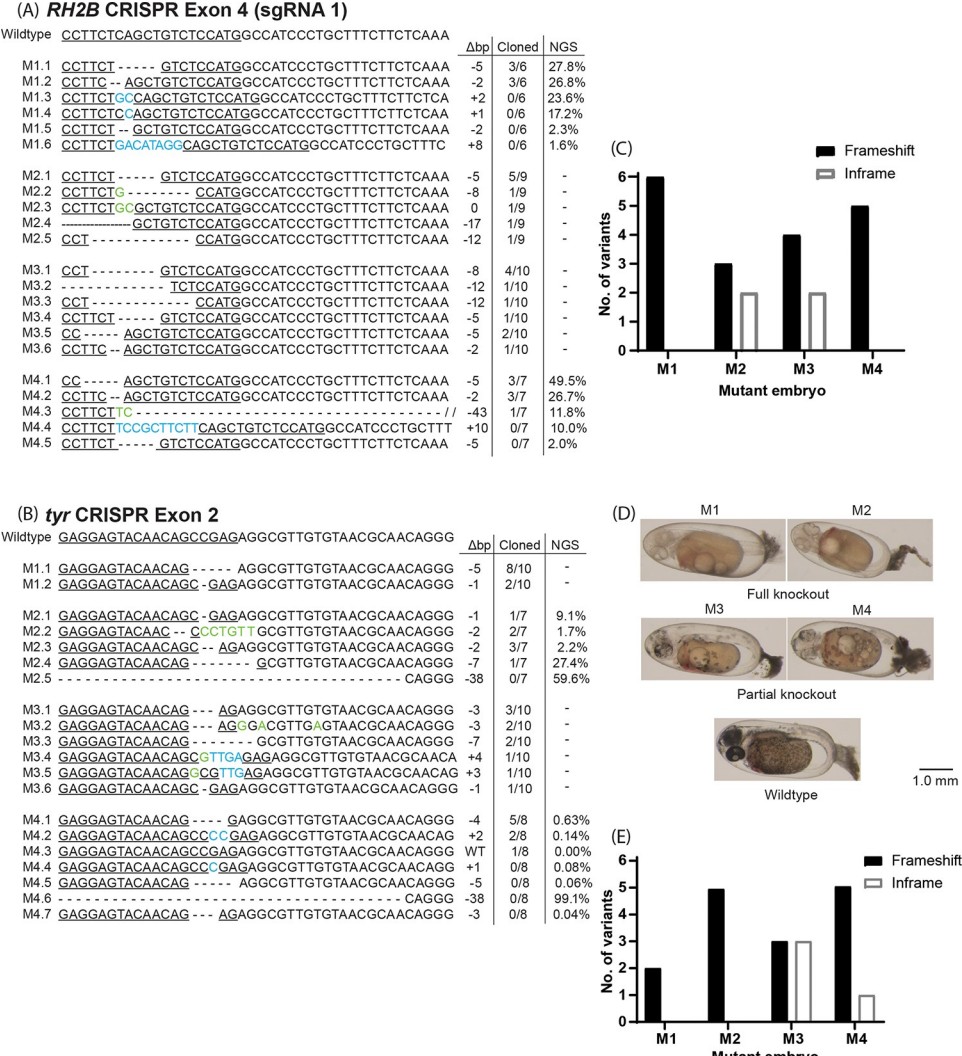

**Fig 3. Genotype analysis of *RH2B* mutant *Amphiprion ocellaris* embryos.** Subcloned sequences and next generation shotgun amplicon sequences (NGS) belonging to *A. ocellaris* embryos (clutch 3, sgRNA *RH2B* 1; clutch 12, sgRNA *tyr* 1) with mutations at targeted sequences (underlined) located on (A) Exon 4 of the *RH2B* opsin gene, and (B) Exon 2 of the *tyr* gene. Wildtype (WT) sequences are included for reference. Mutations included deletions (dashes), substitutions (green), and insertions (blue). Sequence labels on the left-side indicate mutant embryo and allele no., while numbers on the right-side indicate the base pair change (Δbp), proportion of each allele out of the total number of cloned sequences for each embryo, or the percentage (%) of reads out of total reads for NGS. (C) Number of frameshift and in-frame mutations per *RH2B* mutant embryo. (D) Micrographs of *tyr* mutant *A. ocellaris* embryos exhibiting full knockout (*tyr*-M1 and -M2) and partial knockout (*tyr*-M3 and -M4) phenotypes, and a wildtype embryo for comparison. (E) Number of frameshift and in-frame mutations per *tyr* mutant embryo.

mutations. This high number of mutations per embryo suggests Cas9 cutting activity persisted beyond initial cell division, an indication of a high dosage of sgRNA and Cas9, that could potentially be reduced further if desired.

Analysis of the subcloned sequences of *RH2B* mutant juveniles (from clutch 11, *RH2B* 4; Fig 4A), showed biallelic mutations in two out of the three fish examined (Fig 4B). Only one juvenile (M7) was found to possess a WT allele, along with two in-frame mutations (Fig 4A and 4C). Both juveniles M5 and M6, were found to possess only frameshifted sequences

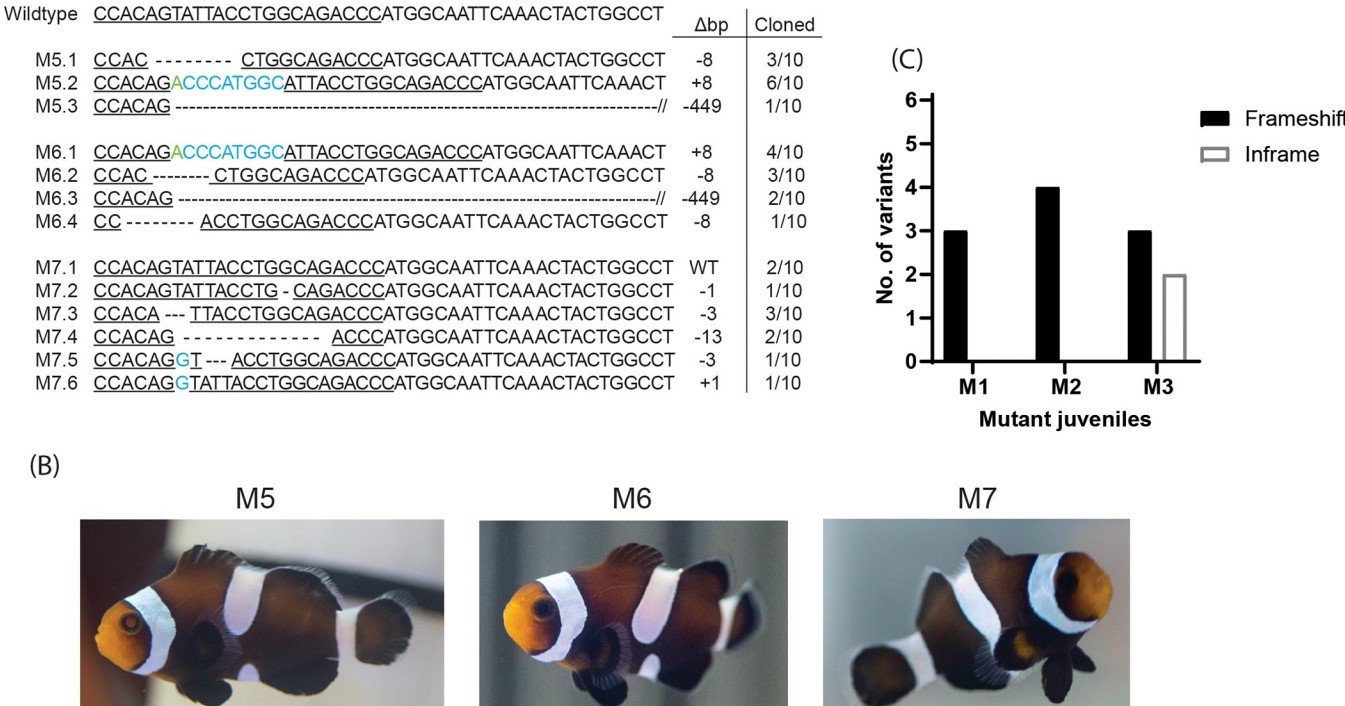

**Fig 4. Genotype analysis of four-month-old *RH2B* mutant *Amphiprion ocellaris*.** (A) Subcloned sequences belonging to *A. ocellaris* juveniles (clutch 11, sgRNA *RH2B* 4) with mutations at targeted sequences (underlined) located on Exon 1 of the *RH2B* opsin gene. Wildtype (WT) sequence is included for reference. Mutations included deletions (dashes), substitutions (green), and insertions (blue). Sequence labels on the left-side indicate mutant fish and allele no., while numbers on the right-side indicate the base pair change (Δbp) and the proportion of each allele out of the total number of cloned sequences for each fish. (B) Images of the *RH2B* mutant *A. ocellaris* juveniles. (C) Number of frameshift and in-frame mutations per *RH2B* mutant fish.

(Fig 4C), which also likely have impaired *RH2B* gene function. This further demonstrated the long-term viability of mutants produced using our CRISPR/Cas9 approach.

Interestingly, analysis of NGS data revealed an identical 38bp deletion in both examined *tyr* mutants (*tyr*-M2.5 and *tyr*-M4.6; Fig 3B), which spanned the entirety of the CRISPR site and PAM. This mutation was found by the majority of mapped reads in mutants *tyr*-M2 (59.6%; n reads = 49608) and *tyr*-M4 (99.1%; n reads = 658075), that is highly unusual when considering that it went completely undetected by the subcloned sequencing analysis of both embryos. Moreover, the location of this mutation is atypical of double-stranded breaks induced by CRISPR/Cas9 [52]. A second NGS run returned similar results, and therefore, it did not appear to be a sequencing or library preparation error. We suggest this deletion was possibly an artefact from PCR during sample preparation, rather than a genuine mutation.

Because there were no easily discernible phenotype(s) in *RH2B* mutant embryos, we speculate on the loss of gene function based on the frameshift or in-frame nature of mutations (Figs 3C and 4C). Four of the seven subcloned *RH2B* mutants (*RH2B*-M1, -M4, -M5, -M6) possessed a full complement of mutant alleles that exhibited frameshifts (Figs 3 and 4). Examination of the translated (frameshifted) sequences (Supporting Information S6 File for an alignment of translated sequences) confirmed the presence of missense mutations that disrupted the chromophore binding site (Lys296), and downstream premature stop codons that may preclude visual pigment formation. Thus, it is likely these four embryos and fish had/have

either a complete knockout or at least impaired *RH2B* gene function. Future attempts to remove the entire chromophore binding site could involve co-injecting upstream and down-stream positioned sgRNA.

## Phenotype analysis of mutants

CRISPR/Cas9 knockout of *A. ocellaris tyr* produced embryos which exhibited varying degrees of hypomelanism (Fig 3D), a phenotype attributed to the disruption of the enzymatic conver-sion of tyrosine into melanin and is similarly observed in *tyr* knockout zebrafish embryos and larvae [53,54]. In comparison, wildtype *A. ocellaris* embryos consistently had heavily pig-mented skin and eyes. A complete lack of melanin was observed in two (*tyr*-M1 and *tyr*-M2) out of the 14 injected embryos from clutch 12 (Fig 3D). Analysis of their subcloned sequences and NGS data revealed both had biallelic mutations, all of which are likely to induce frame-shifts that render TYR non-functional (Fig 3E). Whereas partial depigmentation or a mosaic appearance was found in five out of the 14 embryos (e.g., *tyr*-M3 and *tyr*-M4; Fig 3D), most likely as a result of an incomplete knockout of TYR activity caused by in-frame mutations (*tyr*-M3.1, 3.2, 3.5, and *tyr*-M4.7; Fig 3B and 3D) and/or wild type alleles (*tyr*-M4.3; Fig 3B). The nature of this skin pigmentation phenotype has been shown in zebrafish to be sgRNA/Cas9 dose- dependent [54]; however, in our case the nature of the mutation (i.e., in-frame or out-of-frame) was also a major determinant of phenotype. Notably, no WT allele was detected by NGS in *tyr*-M4 despite being found as a subcloned sequence (*tyr*-M4.3; Fig 3B), it is unclear what may have caused this discrepancy.

Behavioural experiments will be necessary to demonstrate a functional loss of visual opsin in *RH2B* mutant anemonefish, as has been demonstrated in opsin knockout strains of medaka that exhibit impaired spectral sensitivity in optomotor tests [55] and/or altered social behav-iour [56,57]. Applying this same approach to other visual opsin genes could also help attribute the input of different visual pigments to vision (e.g., in colour and/or brightness perception). Similarly, the loss of TYR could also be assessed for its impact on colour sensitivity, as has been reported in zebrafish [58].

## Conclusions and further directions

Here we present the first use of the CRISPR/Cas9 system in a reef fish. Targeting the coding regions of the *RH2B* opsin and *tyr* genes successfully induced indel mutations in up to 73.3% of *A. ocellaris* embryos. Moreover, the analysis of subcloned sequences showed our gene-edit-ing approach was able to produce biallelic mutations with an extremely high efficiency of ~90%, causing loss-of-function mutations in a substantial proportion of F0 *tyr* mutants. Our proven application of this technology greatly facilitates the use of CRISPR/Cas9 for a variety of other genetic applications including making precise (knock-in) gene insertions in anemone-fish; however, this would require significant modification of the sgRNA to utilise homologous recombination or alternative strategies [59]. The precision of both gene knock-in and knock-outs using CRISPR/Cas9 in anemonefishes could possibly benefit from applying microhomol-ogy-mediated end-joining (MMEJ) to exploit short microhomologies flanking a target site to more precisely direct cutting activity [40,60]. Combining our protocol with the latest advance-ments in anemonefish egg-care and larval rearing techniques [40,43], will be key in improving survival to study genome-editing in adult anemonefish. Regardless, this raises an exciting future prospect of conducting genome-editing in *A. ocellaris* to study the genetic basis of vari-ous unique traits in a reef fish.

## Supporting information

**S1 File. sgRNA sequences.** List of injected sgRNA sequences.
(DOCX)

**S2 File. PCR details and sgRNA in-vitro assay.** Primer sequences and PCR routine, and in-vitro cutting assay reagents and incubation steps.
(DOCX)

**S3 File. Yamamoto's ringer's solution.** List of reagents and quantities used to make a salt-balanced solution for eggs.
(DOCX)

**S4 File. Injection parameters and configuration.** Microcapillary settings and pneumatic microinjector settings.
(DOCX)

**S5 File. Full summary of NGS data.**
(DOCX)

**S6 File. Translated alignment of *RH2B* sequences.** Translated sequence alignment of frame-shifted alleles found in RH2B-M1 and -M4, and wildtype (WT) RH2B for reference. Sequences were aligned against bovine rhodopsin (RH1) (NCBI accession no. NP_001014890.1), as an opsin template. The chromophore binding site (bovine RH1 AA no., Lys296) is boxed in blue. Amino acid (AA) numbering schemes were according to WT RH2B (upper) and bovine RH1 (lower). Translated sequences were aligned using MAFFT Alignment (v7.450) in Geneious. MAFFT reference: Katoh, K., & Standley, D. M. (2013). MAFFT Multiple Sequence Alignment Software Version 7: Improvements in Performance and Usability. Molecular Biology and Evolution, 30(4), 772–780. https://doi.org/10.1093/molbev/mst010.
(DOCX)

**S1 Raw images.**
(PDF)

## Acknowledgments

We thank the University of Queensland Biological Resources Aquatics Team, particularly Gillian Lawrence and Gerard Pattison for their support in maintaining marine aquaria and sourcing injection equipment.

## Author Contributions

**Conceptualization:** Laurie J. Mitchell, Justin S. Rhodes, N. Justin Marshall, Karen L. Cheney, Fabio Cortesi.

**Data curation:** Laurie J. Mitchell.

**Formal analysis:** Laurie J. Mitchell, Valerio Tettamanti.

**Funding acquisition:** N. Justin Marshall, Karen L. Cheney, Fabio Cortesi.

**Investigation:** Laurie J. Mitchell, Valerio Tettamanti, Fabio Cortesi.

**Methodology:** Laurie J. Mitchell, Valerio Tettamanti, Justin S. Rhodes, Fabio Cortesi.

**Supervision:** N. Justin Marshall, Karen L. Cheney, Fabio Cortesi.

**Validation:** Laurie J. Mitchell.

**Visualization:** Laurie J. Mitchell.

**Writing – original draft:** Laurie J. Mitchell.

**Writing – review & editing:** Laurie J. Mitchell, Valerio Tettamanti, N. Justin Marshall, Karen L. Cheney, Fabio Cortesi.

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
