## [Decision Letter · Decision Letter 0]

13 May 2021

PONE-D-21-10870

CRISPR/Cas9-mediated generation of biallelic G0 anemonefish (*Amphiprion ocellaris*) mutant embryos

PLOS ONE

Dear Dr. Mitchell,

Thank you for submitting your manuscript to PLOS ONE. After careful consideration, we feel that it has merit but does not fully meet PLOS ONE’s publication criteria as it currently stands. Therefore, we invite you to submit a revised version of the manuscript that addresses the points raised during the review process.

The study is well planned. Authors have to provide more detailed methodology and also validation of the successful knockouts, which are very much critical for this kind of manuscripts.

We look forward to receiving your revised manuscript.

Kind regards,

Rajakumar Anbazhagan, Ph. D.

Academic Editor

PLOS ONE

Journal Requirements:

2.PLOS ONE now requires that authors provide the original uncropped and unadjusted images underlying all blot or gel results reported in a submission’s figures or Supporting Information files. This policy and the journal’s other requirements for blot/gel reporting and figure preparation are described in detail at https://journals.plos.org/plosone/s/figures#loc-blot-and-gel-reporting-requirements and https://journals.plos.org/plosone/s/figures#loc-preparing-figures-from-image-files. When you submit your revised manuscript, please ensure that your figures adhere fully to these guidelines and provide the original underlying images for all blot or gel data reported in your submission. See the following link for instructions on providing the original image data: https://journals.plos.org/plosone/s/figures#loc-original-images-for-blots-and-gels.

Additional Editor Comments:

Authors have to provide more detailed methodology and also validation of the successful knockouts.

Reviewers' comments:

Reviewer's Responses to Questions

**Comments to the Author**

1. Is the manuscript technically sound, and do the data support the conclusions?

Reviewer #1: Partly

Reviewer #2: Partly

Reviewer #3: Yes

2. Has the statistical analysis been performed appropriately and rigorously? 

Reviewer #1: Yes

Reviewer #2: No

Reviewer #3: Yes

3. Have the authors made all data underlying the findings in their manuscript fully available?

Reviewer #1: Yes

Reviewer #2: Yes

Reviewer #3: Yes

4. Is the manuscript presented in an intelligible fashion and written in standard English?

Reviewer #1: Yes

Reviewer #2: No

Reviewer #3: Yes

5. Review Comments to the Author

Reviewer #1: The manuscript by Mitchell et al. explore the potential application of CRISPR/Cas9 targeting in Amphiprion occellaris (anemonefish). The authors demonstrate the successful targeting of rhodopsin-like 2B opsin encoding gene (RH2B), involved in vison and Tyrosinase encoding gene (tyr), involved in the production of melanin. They show clear loss of function in tyr mutant embryos and associated phenotype. Based on this the authors suggest that CRISPR/Cas9 tool can be used for gene-editing in anemonefish, thereby it would help in studying gene function. Overall its straightforward methodology study, suggesting the use of CRISPR/Cas9 application in studying gene function in anemonefish. However, the study lacks functional and behavioral studies of mutant anemonefish fish. Further, CRISPR/Cas9 system has already been used in several fish models, such as zebrafish, tilapia, and medaka (27130213, 24728957) to study gene function or to target a specific gene. As it stands—the authors should include the functional and behavioral study of mutant fish otherwise it just an adaption of the protocol in Amphiprion occellaris and offer no technical advancement over already published studies or provide new insight.

a. The authors suggest that the mortality of experimental eggs dependent on the injection process rather than sgRNA/Cas9 cytotoxicity, however, this hypothesis is not validated.

b. It’s unclear from the study if the off-target analysis is performed.

c. The authors should verify germline transmission mutant embryo as it's important to investigate long-term function study.

d. The titration of sgRNA/Cas9 doses should be performed to optimize cas9 cutting activity.

e. The authors should perform a behavioral and functional study of mutant Amphiprion occellaris to assess the long-term effect of mutation on the viability of the fish.

Reviewer #2: Reviewer Comments to Author(s):

The manuscript describes the use of CRISPR/Cas9 for genome editing in reef fish or anemonefish. While CRISPR/Cas9 mediated mutagenesis has been performed in a previous publication (1), the rational is that no protocol has detailed how to inject anemonefish to generate mosaic mutants or complete knockouts in the F0 generation.

In the manuscript (PONE-D-21-10870), the authors established a system for implementing CRISPR/Cas9 to produce genome edited reef fish. Although the authors microinjection technique affected the survivability of injected embryos, the overall success rate of CRISPR/Cas9 mediated mutagenesis was measurable. Heritability of CRISPR/Cas9 induced mutations were not detected in the injected reef fish due to the model systems long generation time constraints. However, the authors generate mosaic founder (F0) embryos that closely resemble true null mutations, at least in the case of tyr. The ability to generate F0 mutants will allow testing the functional importance of genes, although off-target effect is more common with this approach. The research presented provides the groundwork for CRISPR/Cas9 application in anemonefish.

The methods are appropriate, and the results are clear with exceptions. An overarching concern is the claim that efficient knockdown is achieved to generate a complete knockout in the F0 without clearly explaining the mutational analysis. Overall, the manuscript readability can be revised to improve the flow of the text, but there is sufficient information presented for readers to follow the rationale, procedures and glean insights into the application of the CRISPR/Cas9 system for use on reef fish and other model organisms with longer generation times.

Injected embryos are often referred to as founders or F0 and not G0.

Lines 32, 36, 103 – “Eggs” is often used to refer to one-cell stage “embryos”. In the literature, injected “eggs” and one-cell stage “eggs” are typically “embryos”.

Lines 87, 159, 161. The authors should clarify if CRISPR/Cas9 “constructs” are injected into embryos or CRISPR/Cas9 “components” such as RNA (sgRNA) and protein (Cas9).

Lines 147 – 157 Parts of text can be revised to improve the flow of the article. For example, this section contains text that can be incorporated into the results section.

Clarification regarding the generation of “knockouts” is needed. To make the claim that efficient mutagenesis is achieved to analyze the F0 for functional analysis, it must be clear how reliable the knockdown is.

In the methods section, the authors write that “all positive mutants were heterozygous” (lines 203-204). The authors go on to state that “only one tyr mutant retained a wild-type allele” (line 281) in the results section. Were positive mutants heterozygous or did the authors find that the embryos lacked wild-type sequences when assaying their sequence clones from whole embryos? What is the authors definition of “heterozygous”?

To get a real sense of how mosaic the F0s are, more than a dozen to one hundred sequences per fish should be analyzed. More clarification is needed in regard to how many clones were sequenced (See comments for lines 280 – 282 below).

Line 242 – 247, 260 – 269. Parts of text can be revised to improve the flow of the article. Some text may be moved and incorporated in the discussion section.

Line 280 – 282

The authors should include more information to justify their choice of methods and clearly demonstrate how reliable the knockdown is. I presume the authors selected the RH2B sgRNA target 1 clutch 3 embryos because they see high somatic activity (4/13, 30.8%). Is this the case?

From the 4 embryos subjected to PCR screening for the RH2B gene, those PCR amplicons were subcloned and how many clones were sequenced? Figure 3A would suggest maybe 16 clones were sequenced. Is this also the case for tyr?

According to the text, authors subcloned and sequenced tyr PCR amplicons from embryos that came from clutch 9 (line 279). However, Figure 3 described subcloned sequenced from clutch 8 (lines 316, 331).

In regard to line 308, if the data positions the authors to do so, they can write that “no wild-type sequences were detected in any of the sequenced clones (n=# of sequenced clones).” However, lines 203-204 and 281 suggest the authors may be observing compound heterozygous fish.

Line 292 – 294. A graphical presentation can be made for the indels generated. In a bar graph, a column may show how many in frame mutations and frameshift mutations were made that were deletions and another column of the insertions that were in frame or frameshift mutations.

Another type of graph to depict the distribution of CRISPR/Cas9 induced mutations is to show number of mutations on the y axis and the base pair changes (- and +) plotted on the x axis. In this graph, authors will have bars that indicate deletion range and bars that indicate insertion sizes.

Figure 2

In Figure 2C and Figure 2D, labels for which fragment is uncut and which are cleaved can be presented. The in vitro experiment is successful, but another control is to run the sgRNA alone. In Figure 2C, the large overexposed bands may be tyr1 and tyr2 sgRNAs and not cut DNA.

References:

1. Salis P, Lorin T, Lewis V, Rey C, Marcionetti A, Escande ML, Roux N, Besseau L, Salamin N, Sémon M, Parichy D, Volff JN, Laudet V. Developmental and comparative transcriptomic identification of iridophore contribution to white barring in clownfish. Pigment Cell Melanoma Res. 2019 May;32(3):391-402. doi: 10.1111/pcmr.12766. Epub 2019 Jan 29. PMID: 30633441; PMCID: PMC6483885.

Reviewer #3: The manuscript by Mitchell LJ et al. describes CRISPR-Cas 9 genome editing in anemonefish. The authors efficiently demonstrate successful editing of two separate target sites- RH2B and Tyrosinase producing gene at G0 of A.ocellaris embryos. The experiments are well designed, performed with appropriate controls and the manuscript is well written. The manuscript will be potentially important to study gene functions in other reef fishes.

Comments:

1. Can authors comment on how is the CRISPR-Cas 9 protocol described here is novel and different from previously published CRISPR-Cas 9 editing done in other fishes?

2. Can this protocol be applied to precisely delete specific nucleotides at the target site and also Knock-in specific genes?

3. The authors show loss of function phenotype in tyr KO embryos, is there a functional phenotype to demonstrate loss of R2BH in the embryos? Probably by Immunohistochemical analysis of opsin trafficking in photoreceptors?

4. Comments on any off-target/unpredicted mutations.

6. PLOS authors have the option to publish the peer review history of their article (what does this mean?). If published, this will include your full peer review and any attached files.

Reviewer #1: No

Reviewer #2: No

Reviewer #3: No

---

## [Author Response · Author response to Decision Letter 0]

13 Nov 2021

Response to the editor and reviewers

We would like to thank Dr Anbazhagan and the three anonymous reviewers for taking the time to review our manuscript and provide insightful feedback.

We have now addressed all the concerns that were raised, and a point-by-point reply can be found below. Two versions of the manuscript, one with track-changes and one without have also been uploaded. 

We hope our revised manuscript meets the requirements for publication in PLOS ONE and look forward to your decision. 

Yours sincerely,

Laurie Mitchell

On behalf of all authors. 

Additional Editor Comments:

Authors have to provide more detailed methodology and also validation of the successful knockouts.

- These concerns have also been raised by the reviewers and we provide detailed answers about how we addressed them below.

Reviewer #1:

Reviewer #1: The manuscript by Mitchell et al. explore the potential application of CRISPR/Cas9 targeting in Amphiprion occellaris (anemonefish). The authors demonstrate the successful targeting of rhodopsin-like 2B opsin encoding gene (RH2B), involved in vison and Tyrosinase encoding gene (tyr), involved in the production of melanin. They show clear loss of function in tyr mutant embryos and associated phenotype. Based on this the authors suggest that CRISPR/Cas9 tool can be used for gene-editing in anemonefish, thereby it would help in studying gene function. Overall its straightforward methodology study, suggesting the use of CRISPR/Cas9 application in studying gene function in anemonefish. However, the study lacks functional and behavioral studies of mutant anemonefish fish. Further, CRISPR/Cas9 system has already been used in several fish models, such as zebrafish, tilapia, and medaka (27130213, 24728957) to study gene function or to target a specific gene. As it stands—the authors should include the functional and behavioral study of mutant fish otherwise it just an adaption of the protocol in Amphiprion occellaris and offer no technical advancement over already published studies or provide new insight.

- The reviewer is concerned that our manuscript provides no technical advancement over already published studies using other teleosts with the CRISPR/Cas9 platform and requested for a behaviour or functional characterisation of the targeted genes in our study. 

Regarding the first concern, we understand where the reviewer is coming from in that we here present a protocol to use CRISPR/Cas9 in another fish species. However, as opposed to the classical model systems for which the technology was adapted and applied in previously (e.g., medaka, zebrafish, and tilapia), reverse genetics in coral reef fishes faces several major challenges and to the best of our knowledge, has not been attempted before. The biggest challenge is that with demersal spawning fishes such as many smaller coral reef fishes, the eggs are attached to the substrate. For that reason, we had to develop a different approach for microinjection all together i.e., injecting eggs/embryos on shards of clay pots while attached as opposed to injecting into loose eggs/embryos as done in other fish species (lines 123-125, 211-212). The chorion of anemonefish eggs also behaves different to the one found in e.g., zebrafish, and thus needle dimensions had to be adjusted. Likewise, other injection parameters such as injection pressure etc., as well as concentrations of the vector had to be adjusted. The next major challenge is to maintain the embryos, hatching and rearing of the fry. Demersal spawners such as Amphiprion ocellaris provide egg care i.e., they clean the eggs from debris, remove dead eggs, aerate them, and protect them from predation. However, injected eggs/embryos are rejected by the parents and hence, we had to develop a method to substitute parental care. This new husbandry approach is documented in detail in lines 128-140. Hence, like the CRISPR/Cas9 protocols that have been and continue to be published for other non-model organisms e.g., corals, axolotl, fathead minnow, our approach includes major advancements that warrants separate publication. 

Regarding the functional characterisation of mutant fishes. We now provide details on fully developed (4-month post-hatch), healthy RH2B opsin mutant fish in our care that have been successfully reared since our initial submission (lines 374-379, Figure 4). Sanger sequencing performed on subcloned sequences of four (out of seven) mutants shows that two are full knockouts and one is a partial knockout containing the wildtype allele (Figure 4). While a full behavioural experiment (e.g., tests for colour vision deficiency) would be ideal, this is beyond the scope of the current study. The purpose of this study is to provide, for the first time, a detailed protocol for CRISPR/Cas9 manipulation in a non-model coral reef fish species. Extensive behavioural studies take considerable amount of time, and we plan to conduct such experiments in a standalone study. 

We also used amplicon shot-gun sequencing on previously extracted mutant embryos (2x RH2B mutants and 2x Tyr mutants) to gain a more complete picture of the mutations present (lines 341-348; Figure 3 A, B). This shows that most mutant alleles were picked up with the cloning approach. Most importantly, full-knockout individuals as determined by cloning did not show any wild-type alleles when using this vastly increased sequencing depth. We are therefore confident that full-knockout F0 anemonefishes can be produced using our approach. 

a. The authors suggest that the mortality of experimental eggs dependent on the injection process rather than sgRNA/Cas9 cytotoxicity, however, this hypothesis is not validated.

- We have since added survival rate data from further injection rounds that used a lower concentration of sgRNA and Cas9 (lines 231, 307-309, Table 1). This found no clear improvement in survival despite maintaining a high efficiency in mutation, suggesting mortality was mainly attributable to physical trauma rather than sgRNA/Cas9 cytotoxicity.

b. It’s unclear from the study if the off-target analysis is performed.

- Our methods section (lines 172-182) describes our off-target analysis when designing our sgRNA sequences, which takes advantage of the high-resolution genome assembly for this species. We have now added further detail on this off-target screening analysis. This used an inbuilt off-target screen in the ‘Find CRISPR sites’ function of the software ‘Geneious’. The application offers a number of candidate-sgRNA sequences at a predefined region of interest for the user to select. As part of the metrics given for each sgRNA there is a specificity score calculated using a renowned method developed by the Zhang Lab at MIT.

Taken from the Geneious webpage:

“Each off-target site is given a score based on how similar it is to the original CRISPR site and where any mismatches occur (mismatches near the PAM site will affect binding more than mismatches further away from the PAM site). A higher score for an off-target site indicates a higher similarity to the original CRISPR site (and thus a higher likelihood of the CRISPR/Cas complex binding to the off target). The overall specificity score for a CRISPR site is 100% minus a weighted sum of off-target scores in the target genome. Thus, a higher specificity score indicates a better CRISPR site with few or weak potential offsite targets.”

(See https://www.geneious.com/tutorials/finding-crispr-sites-r9/) 

Our database scored against the assembled genome of A. ocellaris, and we only selected sgRNAs with a score of >90%, where our RH2B sgRNAs had specificity ranging from 90 – 100%, while the tyr sgRNAs both scored 100%.

Although we cannot be absolutely certain of no off-target activity, it seems extremely unlikely. Our sgRNAs certainly induced mutations at the desired locations as verified via sequencing and also phenotypic scoring for tyr mutants.

[for full details on the specificity method see Hsu et al. (2013). DNA targeting specificity of RNA-guided Cas9 nucleases. Nat Biotechnol.].

The authors should verify germline transmission mutant embryo as it's important to investigate long-term function study.

- While we recognise the importance of analysing germline transmission for long-term experiments (i.e., for establishing homozygous lines). In our case, we want to stress the importance of CRISPR/Cas9 as a means of performing rapid gene-editing experiments to ascertain gene function within the first generation of mutants and eliminating the need for multigenerational breeding schemes. Unlike model systems (e.g., zebrafish, medaka, mice), the generation time of many non-model animals prohibits long-term experiments, and as such establishing a stable germline before starting experiments is often not feasible. Anemonefish typically take ~9-18 months to reach full sexual maturity and form a stable breeding pair, whereby the males reach sexual maturity first and females usually take a minimum of 12 months to produce their first clutch. To give an idea, this would require a minimum of 36 months to achieve a single F3 screening scheme with anemonefish. One of the advantages of the CRISPR/Cas9 platform is the ability to produce knockout mutations within the first generation of individuals and does not necessitate breeding (see e.g., Fei, J.F., et la., 2018. Application and optimization of CRISPR–Cas9-mediated genome engineering in axolotl (Ambystoma mexicanum). Nature protocols, 13(12), pp.2908-2943.). This is one of the major reasons for our development of this protocol with anemonefish, as this will enable others to perform gene-editing experiments without the need for extensive breeding schemes. To emphasise this point more strongly, we have now included a sentence explicitly stating the long generation time of anemonefish making them unsuitable for gene-editing experiments which require breeding schemes (lines 88-90). In the mid- to long-term we aim at producing stable lines for mutants of special interest to be shared with the community.

d. The titration of sgRNA/Cas9 doses should be performed to optimize cas9 cutting activity.

- Please see our answer to ‘a.’ on line 231 we describe another injection treatment using a reduced 5 µm sgRNA/Cas9 concentration, and then show that cutting efficiency was maintained (Table 1 clutches 8-11, and lines 281-283). While our current cutting efficiency is more than adequate to produce full-knockout progeny, we suggest other potential options that may improve it further (see lines 309-320).

e. The authors should perform a behavioral and functional study of mutant Amphiprion occellaris to assess the long-term effect of mutation on the viability of the fish.

- Please see our initial response. In short, while a full behavioural experiment (e.g., tests for colour vision deficiency) would be ideal, this is beyond the scope of the current study which is a protocol, as a proof-of-concept for performing CRISPR/Cas9 in a reef fish. Extensive behavioural studies take considerable amount of time, and we plan to conduct this as an entirely separate study/experiment. However, we have now included sequencing results for (three-month-old) juvenile mutants successfully raised in our hatchery system. This shows the long-term viability of our mutagenic fish.

Reviewer #2

Reviewer #2: Reviewer Comments to Author(s):

The manuscript describes the use of CRISPR/Cas9 for genome editing in reef fish or anemonefish. While CRISPR/Cas9 mediated mutagenesis has been performed in a previous publication (1), the rational is that no protocol has detailed how to inject anemonefish to generate mosaic mutants or complete knockouts in the F0 generation.

- CRISPR/Cas9 has not been attempted in anemonefishes or reef fishes before. The study the reviewer is referring to here, used zebrafish to test orthologous gene sequences for iridophore development that were originally detected in anemonefishes using a comparative transcriptomic approach.

In the manuscript (PONE-D-21-10870), the authors established a system for implementing CRISPR/Cas9 to produce genome edited reef fish. Although the authors microinjection technique affected the survivability of injected embryos, the overall success rate of CRISPR/Cas9 mediated mutagenesis was measurable. Heritability of CRISPR/Cas9 induced mutations were not detected in the injected reef fish due to the model systems long generation time constraints. However, the authors generate mosaic founder (F0) embryos that closely resemble true null mutations, at least in the case of tyr. The ability to generate F0 mutants will allow testing the functional importance of genes, although off-target effect is more common with this approach. The research presented provides the groundwork for CRISPR/Cas9 application in anemonefish.

The methods are appropriate, and the results are clear with exceptions. An overarching concern is the claim that efficient knockdown is achieved to generate a complete knockout in the F0 without clearly explaining the mutational analysis. Overall, the manuscript readability can be revised to improve the flow of the text, but there is sufficient information presented for readers to follow the rationale, procedures and glean insights into the application of the CRISPR/Cas9 system for use on reef fish and other model organisms with longer generation times.

- We recognise the need to make our mutation analysis clearer (see below for specific addressment).

Injected embryos are often referred to as founders or F0 and not G0.

- We have amended our manuscript to now refer to all injected embryos as founders or ‘F0’.

Lines 32, 36, 103 – “Eggs” is often used to refer to one-cell stage “embryos”. In the literature, injected “eggs” and one-cell stage “eggs” are typically “embryos”.

- All relevant lines have now been amended to read “embryos” for the post-fertilised, single cell stage.

Lines 87, 159, 161. The authors should clarify if CRISPR/Cas9 “constructs” are injected into embryos or CRISPR/Cas9 “components” such as RNA (sgRNA) and protein (Cas9).”

- We have now clarified in the manuscript that sgRNA/Cas9 complexes are injected. This complex is formed during the incubation period that precedes injecting (see line 222)

Lines 147 – 157 Parts of text can be revised to improve the flow of the article. For example, this section contains text that can be incorporated into the results section.

- We have moved the mention of using two guides to cut out the entire gene region to the results/discussion section (see lines 407-409). The remaining text in this section pertains to our sgRNA design and as such should remain in the methods.

Clarification regarding the generation of “knockouts” is needed. To make the claim that efficient mutagenesis is achieved to analyze the F0 for functional analysis, it must be clear how reliable the knockdown is.

In the methods section, the authors write that “all positive mutants were heterozygous” (lines 203-204). The authors go on to state that “only one tyr mutant retained a wild-type allele” (line 281) in the results section. Were positive mutants heterozygous or did the authors find that the embryos lacked wild-type sequences when assaying their sequence clones from whole embryos? What is the authors definition of “heterozygous”?

- We now recognise that our previous use of terminology was inconsistent and confusing. Now, we have amended throughout the manuscript that heterozygous mutants are those with two or more distinct alleles, as opposed to homozygous (i.e., one allele type present). In our case, all the embryos/fish are heterozygous, but only one individual contained a wild-type allele. None of our individuals showed only a single mutation. Furthermore, our NGS results for two RH2B mutants and two tyr mutants found no evidence of wildtype alleles, indicating that our knockout was reliable.

To get a real sense of how mosaic the F0s are, more than a dozen to one hundred sequences per fish should be analyzed. More clarification is needed in regard to how many clones were sequenced (See comments for lines 280 – 282 below).

We have now also provided NGS amplicon sequencing results for the two individuals (per target gene) which had the least number of subcloned sequences i.e., we sequenced hundreds of thousands to millions of sequences per individual. Variant sequences only detected via NGS are enclosed in boxes and given a percentage out of the total reads that covered the target region. While the most frequent sequences detected were all represented by the subcloning analysis, we did find a few previously undetected variants, but none were WT or represented in-frame mutations. The amplicon sequencing approach confirms that F0 full-knockout individuals can be produced using our method. 

Line 242 – 247, 260 – 269. Parts of text can be revised to improve the flow of the article. Some text may be moved and incorporated in the discussion section.

- The mentioned section is a combined results/discussion. For lines 242-247, the possibility of using a smaller needle-size to possibly reduce mortality follows the results on survivorship. We believe this to be the most appropriate area in the manuscript. Similarly, in lines 260-269, after stating the mutation rate we proceed onto suggesting possible means of improving it. Again, we believe this current structure is a best-fit and cannot find a more appropriate place to move this information. We are open to ideas though in case the reviewer has a specific location in mind.

Line 280 – 282

The authors should include more information to justify their choice of methods and clearly demonstrate how reliable the knockdown is. I presume the authors selected the RH2B sgRNA target 1 clutch 3 embryos because they see high somatic activity (4/13, 30.8%). Is this the case?

From the 4 embryos subjected to PCR screening for the RH2B gene, those PCR amplicons were subcloned and how many clones were sequenced? Figure 3A would suggest maybe 16 clones were sequenced. Is this also the case for tyr?

According to the text, authors subcloned and sequenced tyr PCR amplicons from embryos that came from clutch 9 (line 279). However, Figure 3 described subcloned sequenced from clutch 8 (lines 316, 331).”

- More information on our reasons for choosing the clutches for subcloning has now been provided (lines 246-247). They were indeed chosen as clutches with a high amount of somatic activity. The numbers (‘X/X’) after each sequence in Figure 3 reflects the number of the particular variant sequences detected out of the total sampled subcloned colonies per individual (i.e., M1 = 6, M2 = 9, M3 = 10, and M4 = 7 for RH2B; M1 = 10, M2 = 7, M3 = 10, M4 = 8 for tyr). This has now been more clearly stated in the caption for Figure 3. For RH2B-M1 and -M4, and tyr-M2 and -M4, we have also now performed NGS amplicon sequencing that yielded a few previously undetected variants (see lines 341-348 and sequences in Figure 3 A, B). We have changed the text to reflect our use of tyr clutch 12 in our subcloning analysis. 

Note: what was previously clutch 8 is now clutch 12 due to since adding the RH2B Exon 1 injected clutches (no. 8 to 11).

In regard to line 308, if the data positions the authors to do so, they can write that “no wild-type sequences were detected in any of the sequenced clones (n=# of sequenced clones).” However, lines 203-204 and 281 suggest the authors may be observing compound heterozygous fish.

- We have changed this to read that no wild-type sequences were detected in any of the sequenced clones for RH2B. 

Line 292 – 294. A graphical presentation can be made for the indels generated. In a bar graph, a column may show how many in frame mutations and frameshift mutations were made that were deletions and another column of the insertions that were in frame or frameshift mutations.

Another type of graph to depict the distribution of CRISPR/Cas9 induced mutations is to show number of mutations on the y axis and the base pair changes (- and +) plotted on the x axis. In this graph, authors will have bars that indicate deletion range and bars that indicate insertion sizes.

- We have now added two column graphs (see Figures 3 C and 4 C) showing the number of frameshift and in-frame mutations detected in each individual embryo. Although we have not separated this by insertions and deletions, as this does not add any information to the figure, where the nature and size (+/- Δbp) of the mutations are already stated after each variant sequence.

Figure 2

In Figure 2C and Figure 2D, labels for which fragment is uncut and which are cleaved can be presented. The in vitro experiment is successful, but another control is to run the sgRNA alone. In Figure 2C, the large overexposed bands may be tyr1 and tyr2 sgRNAs and not cut DNA.”

- We have now added boxes around cut fragments. Both target genes have negative controls (either no sgRNA and Cas9), and positive controls (sgRNA and no Cas9 for RH2B, or Cas9 and no sgRNA for tyr). This shows that fragment size differences were due to CRISPR/Cas9 cutting activity. We have since amended the figure to depict the correct positive control for tyr. The lower smeared region very possibly contains sgRNA; however, its persistence in the positive control for tyr (containing no sgRNA) raises doubt and it may simply be remnant primer contained in the PCR product (no clean-up routine was done). In any case, the well-defined fragment (now boxed in the figure) above the smear region cannot be sgRNA, as this fragment exceeds 100bp and our sgRNAs are only 23bp in length. 

References:

1. Salis P, Lorin T, Lewis V, Rey C, Marcionetti A, Escande ML, Roux N, Besseau L, Salamin N, Sémon M, Parichy D, Volff JN, Laudet V. Developmental and comparative transcriptomic identification of iridophore contribution to white barring in clownfish. Pigment Cell Melanoma Res. 2019 May;32(3):391-402. doi: 10.1111/pcmr.12766. Epub 2019 Jan 29. PMID: 30633441; PMCID: PMC6483885.”

Reviewer #3

Reviewer #3: The manuscript by Mitchell LJ et al. describes CRISPR-Cas 9 genome editing in anemonefish. The authors efficiently demonstrate successful editing of two separate target sites- RH2B and Tyrosinase producing gene at G0 of A.ocellaris embryos. The experiments are well designed, performed with appropriate controls and the manuscript is well written. The manuscript will be potentially important to study gene functions in other reef fishes.

“1. Can authors comment on how is the CRISPR-Cas 9 protocol described here is novel and different from previously published CRISPR-Cas 9 editing done in other fishes?”

- Please refer to our detailed answer to reviewer 1. 

2. Can this protocol be applied to precisely delete specific nucleotides at the target site and also Knock-in specific genes?

- Certainly, but this would involve heavily modifying the sgRNA design to perform knock-ins. Such an approach is already used with some success in other organisms [e.g., for a review on zebrafish see Albadri et al. (2017). Genome editing using CRISPR/Cas9-based knock-in approaches in zebrafish]. For an in-frame knock in this commonly requires homologous recombination (rather than non-homologous end joining for knockouts) with the insertion of DNA flanked by homologous sequences or arms on either side of the region of interest. Similarly, to delete specific nucleotides this could be enabled by microhomology mediated end-joining (MMEJ) by using short homologous arms flanking the target sequence. We have added brief mention in our discussion (lines 443-451) for how both these approaches could be introduced to perform other edits using anemonefish.

3. The authors show loss of function phenotype in tyr KO embryos, is there a functional phenotype to demonstrate loss of R2BH in the embryos? Probably by Immunohistochemical analysis of opsin trafficking in photoreceptors?

- We have not performed any functional characterisation for RH2B mutants, as we believe this is beyond the methodologically focused aspect of the paper and more suitable for an extensive experimental paper that will examine in detail the function of different opsins in anemonefish colour vision. Certainly, in-situ hybridisation and/or immunohistochemistry will be part of that future study, as will behavioural testing of colour vision. 

4. Comments on any off-target/unpredicted mutations.

- Our methods section (lines 172-182) describes our off-target analysis when designing our sgRNA sequences. This used an inbuilt off-target screen in the ‘Find CRISPR sites’ function of the software ‘Geneious’. The application offers a number of candidate sgRNA sequences at a predefined region of interest for the user to select. As part of the metrics given for each sgRNA there is a specificity score calculated using a renowned method developed by the Zhang Lab at MIT. Please also refer to our detailed answer to Reviewer 1 on this question.

---

## [Decision Letter · Decision Letter 1]

1 Dec 2021

CRISPR/Cas9-mediated generation of biallelic G0 anemonefish (*Amphiprion ocellaris*) mutants

PONE-D-21-10870R1

Dear Dr. Mitchell,

We’re pleased to inform you that your manuscript has been judged scientifically suitable for publication and will be formally accepted for publication once it meets all outstanding technical requirements.

Kind regards,

Rajakumar Anbazhagan, Ph. D.

Academic Editor

PLOS ONE

Additional Editor Comments (optional):

Reviewers' comments:

Reviewer's Responses to Questions

**Comments to the Author**

1. If the authors have adequately addressed your comments raised in a previous round of review and you feel that this manuscript is now acceptable for publication, you may indicate that here to bypass the “Comments to the Author” section, enter your conflict of interest statement in the “Confidential to Editor” section, and submit your "Accept" recommendation.

Reviewer #1: All comments have been addressed

Reviewer #2: All comments have been addressed

2. Is the manuscript technically sound, and do the data support the conclusions?

Reviewer #1: Yes

Reviewer #2: Yes

3. Has the statistical analysis been performed appropriately and rigorously? 

Reviewer #1: Yes

Reviewer #2: Yes

4. Have the authors made all data underlying the findings in their manuscript fully available?

Reviewer #1: Yes

Reviewer #2: Yes

5. Is the manuscript presented in an intelligible fashion and written in standard English?

Reviewer #1: Yes

Reviewer #2: Yes

6. Review Comments to the Author

Reviewer #1: (No Response)

Reviewer #2: Reviewer Comments to Author(s):

The revised manuscript describes the use of CRISPR/Cas9 for genome editing in reef fish or anemonefish. While CRISPR/Cas9 mediated mutagenesis has been performed in a previous publication (1), the rational is that no protocol has detailed how to inject anemonefish to generate mosaic mutants or complete knockouts in the F0 generation. In the revised manuscript (PONE-D-21-10870_R1), the authors established a system for implementing CRISPR/Cas9 to produce genome edited reef fish. Although the authors microinjection technique affected the survivability of injected embryos, the overall success rate of CRISPR/Cas9 mediated mutagenesis was measurable. Heritability of CRISPR/Cas9 induced mutations were not detected in the injected reef fish due to the model systems long generation time constraints. However, the authors generate mosaic founder (F0) embryos that closely resemble true null mutations, at least in the case of tyr. The ability to generate F0 mutants will allow testing the functional importance of genes, although off-target effect is more common with this approach. The research presented provides the groundwork for CRISPR/Cas9 application in anemonefish and has a greater impact on the advancement of the use of non-traditional model organisms in scientific discovery.

Overall, the authors have addressed all comments, questions and concerns in the revised manuscript. The scientific arguments and interpretation are accurate and consistent with the results presented in the revised manuscript. The revised manuscript readability and flow of the text is presented with sufficient information for readers to follow the rationale, procedures and glean insights into the application of the CRISPR/Cas9 system for use on reef fish and other model organisms with longer generation times.

References:

1. Salis P, Lorin T, Lewis V, Rey C, Marcionetti A, Escande ML, Roux N, Besseau L, Salamin N, Sémon M, Parichy D, Volff JN, Laudet V. Developmental and comparative transcriptomic identification of iridophore contribution to white barring in clownfish. Pigment Cell Melanoma Res. 2019 May;32(3):391-402. doi: 10.1111/pcmr.12766. Epub 2019 Jan 29. PMID: 30633441; PMCID: PMC6483885.

7. PLOS authors have the option to publish the peer review history of their article (what does this mean?). If published, this will include your full peer review and any attached files.

Reviewer #1: No

Reviewer #2: No

---

## [Editor Report · Acceptance letter]

6 Dec 2021

PONE-D-21-10870R1 

CRISPR/Cas9-mediated generation of biallelic F0 anemonefish (*Amphiprion ocellaris*) mutants 

Dear Dr. Mitchell:

I'm pleased to inform you that your manuscript has been deemed suitable for publication in PLOS ONE. Congratulations! Your manuscript is now with our production department. 

Kind regards, 

on behalf of

Dr. Rajakumar Anbazhagan 

Academic Editor

PLOS ONE